# Optimizing Cell Density and Unveiling Cytotoxic Profiles of DMSO and Ethanol in Six Cancer Cell Lines: Experimental and In Silico Insights

**DOI:** 10.3390/mps8040093

**Published:** 2025-08-10

**Authors:** Abutaleb Asiri, Munazzah Tasleem, Muwadah Al Said, Abdulaziz Asiri, Ali Ahmed Al Qarni, Ahmed Bakillah

**Affiliations:** 1King Abdullah International Medical Research Center (KAIMRC), Eastern Region, Al Ahsa 36428, Saudi Arabia; asiriabu@kaimrc.edu.sa (A.A.); saidm@kaimrc.edu.sa (M.A.S.); qarniaa@mngha.med.sa (A.A.A.Q.); 2Biomedical Research Core, King Saud bin Abdulaziz University for Health Sciences (KSAU-HS), Al Ahsa 36428, Saudi Arabia; 3King Abdulaziz Hospital, Ministry of National Guard-Health Affairs (MNG-HA), Al Ahsa 36428, Saudi Arabia; 4Department of Public Health, College of Applied Medical Sciences in Al-Namas, University of Bisha, Al-Namas City 67392, Saudi Arabia; mtaslem@ub.edu.sa; 5Department of Medical Laboratory Sciences, College of Applied Medical Sciences, University of Bisha, 255, Al Nakhil, Bisha 67714, Saudi Arabia; amfasiri@ub.edu.sa

**Keywords:** cancer cell lines, viability, MTT assay, cell density, cytotoxicity, DMSO, ethanol, in silico analysis

## Abstract

**Background:** Accurate assessment of drug cytotoxicity in vitro is essential for preclinical evaluation of anticancer agents. Methodological parameters such as cell density and solvent concentrations can significantly influence the reproducibility and reliability of cell-based assay results. **Objective:** This study aims to optimize cell seeding density and evaluate the cytotoxic effects of common solvents (DMSO and ethanol) on different cancer cell lines, complemented by in silico analysis to elucidate underlying mechanisms. **Materials and Methods:** Six cancer cell lines (HepG2, Huh7, HT29, SW480, MCF-7, and MDA-MB-231) were seeded at different densities to determine the optimal cell seeding number ideal for cell viability assay at 24, 48, and 72 h. The cytotoxicity of DMSO and ethanol was assessed in these cell lines using an MTT assay at multiple time points. In silico docking studies were conducted to investigate the interactions between solvents and key proteins involved in apoptosis, membrane function, and metabolism. **Results:** A cell density of 2000 cells per well yielded consistent linear viability across cell lines and time points. DMSO at 0.3125% showed minimal cytotoxicity across all cell lines (except MCF-7) and time points; the cytotoxic effect at higher concentrations is variable depending on cell type and exposure duration. Ethanol exhibited rapid and concentration-dependent cytotoxicity, reducing viability by more than 30% at as low as 0.3125% concentration after 24 h. Docking analyses revealed that DMSO binds specifically to apoptotic and membrane proteins, suggesting a role in inducing apoptosis. In contrast, ethanol primarily interacts with metabolic proteins, consistent with its effect on membrane disruption and rapid cell death. **Conclusion:** DMSO at 0.3125% is a good choice as a solvent since it has low toxicity in most tested cell lines; however, the safe concentration limit is dependent on cell type and exposure duration. Ethanol exhibited higher cytotoxicity, necessitating careful concentration management. The in silico analysis supports these findings, indicating that DMSO interacts with apoptosis-related proteins, whereas ethanol primarily affects metabolic processes. These results highlight the importance of precise cell density optimization and solvents for reliable cytotoxicity assessment in cell-based assays.

## 1. Introduction

Cancer remains one of the leading causes of morbidity and mortality worldwide, underscoring the urgent need for novel therapeutic strategies and a deeper understanding of tumor biology [1]. Drug screening in vitro using tumor cell lines is a fundamental step in identifying potential anticancer agents [2,3]. The MTT (3-(4,5-dimethyl-2-thiazolyl)-2,5-diphenyl-tetrazolium bromide) assay is among the various assays widely employed due to its reliability and simplicity in evaluating cell viability [4]. This assay measures the enzymatic reduction of yellow MTT to purple formazan by mitochondrial dehydrogenases in metabolically active cells, serving as an indicator of cell viability [5]. Owing to its widespread use and ability to provide quantitative and reproducible data, this assay is suitable for assessing the impact of compounds on cell viability.

Cell density is a critical factor influencing the accuracy and reproducibility of MTT assays, as it affects cellular metabolism and assay sensitivity [6,7]. Too low a cell density yields weak signals that are difficult to quantify, reducing data reliability [8]. Conversely, excessively high cell density can lead to nutrient depletion, altered cellular behavior, or signal saturation, resulting in inaccurate measurements [7]. Therefore, optimizing cell seeding densities is crucial for generating reliable and reproducible data, thereby facilitating meaningful comparisons across studies. Cancer cell lines such as HepG2 and Huh7 (hepatocellular carcinoma), MCF-7 and MDA-MB-231 (breast cancer), and HT29 and SW480 (colorectal cancer) are commonly used models due to their clinical relevance and distinct biological features [9,10,11]. However, the optimal seeding densities for these cell lines across different time points (24, 48, and 72 h) have not been fully characterized, limiting the reproducibility and comparability of findings.

In drug discovery studies, solvents like Dimethyl Sulfoxide (DMSO) and ethanol are commonly used as vehicles [12]. In many studies, 0.1% (*v*/*v*) DMSO and ethanol in culture medium are commonly employed in cell-based assays to facilitate accurate drug screening [13,14,15,16]. However, these solvents possess intrinsic cytotoxic properties that can confound experimental outcomes by amplifying or mimicking drug effects [17,18]. For example, DMSO has been shown to induce apoptosis by elevating reactive oxygen species (ROS) production and affecting mitochondrial functions, including membrane potential impairment, release of cytochrome c, and activation of enzymes [19,20]. While Ethanol exposure can lead to increased membrane fluidity, changes in membrane proteins composition influencing their functions, and increased levels of oxidative stress [21]. Moreover, different cell lines exhibit variable sensitivities to these solvents, due to cell-specific factors, such as membrane composition and metabolic capacities, which can potentially lead to misleading interpretations [22,23,24,25,26,27]. Despite their widespread use, detailed assessments of the cytotoxic effects of DMSO and ethanol across various cancer cell lines and different cell growth time points are limited, emphasizing the need to establish safe and cell-specific concentration thresholds to improve data reliability and interpretability in drug screening studies. A critical aspect of in vitro cytotoxicity assessment is the interpretation of cell viability data. While statistical significance can detect small differences between groups, it does not necessarily reflect biological relevance [28]. The ISO 10993-5:2009 standard specifies that a reduction in cell viability exceeding 30% relative to the control is indicative of cytotoxicity, serving as a practical threshold for biological significance [29,30]. Relying solely on *p*-value may lead to overestimating minor effects, whereas applying this threshold ensures effects are meaningful in a biological context. Employing an MTT assay alongside this standard provides a robust framework for evaluating solvent cytotoxicity aligned with both regulatory and biological criteria.

Beyond experimental evaluation, understanding the intra-molecular interactions underlying solvent-induced cytotoxicity can provide valuable insights. Therefore, exploring intra-molecular interactions by employing in silico approaches between these solvents and cellular components can elucidate their mechanisms of action at the molecular level in the mentioned cell lines. Such approaches can fill existing gaps in knowledge regarding solvent effects on cell membranes and inform safer experimental designs.

Therefore, we hypothesize that DMSO and ethanol exert dose and time-dependent cytotoxic effects through distinct molecular pathways that vary among different cancer cell lines. Specifically, we propose that DMSO may affect apoptotic and membrane proteins, while ethanol likely disrupts membrane integrity and metabolic processes. This study aims to optimize cell densities, evaluate the cytotoxic effects of DMSO and ethanol at multiple time points, and explore their molecular interactions with key proteins through in silico docking analyses. The findings will enhance the reliability of in vitro screening methods and deepen the understanding of the mechanisms underlying solvent-related cytotoxicity in cancer research.

## 2. Materials and Methods

### 2.1. Cells and Reagents

Six cancer cell lines were used in this study: HepG2 (ATCC ID: HB-8065), Huh7 (Cat. No. T8973), MCF-7 (ATCC ID: HTB-22), MDA-MB-231 (ATCC ID: HTB-26), HT29 (ATCC ID: HTB-38), and SW480 (ATCC ID: CCL-228) were all obtained from ATCC, except the Huh7 cell line, which was obtained from Applied Biological Materials Inc. DMSO was obtained from Thermo Fisher Scientific (Eugene, OR, USA, Cat. No. D12345) and ethanol was obtained from Decon Labs Inc. (King of Prussia, PA, USA, Cat. No. V1001). The MTT assay was from HiMedia (Maharashtra, India, Cat. No. CCK003).

### 2.2. Cell Culture

Cell lines were cultured with Dulbecco’s Modified Eagle Medium (DMEM; Thermo Fisher Scientific, Waltham, MA, USA) supplemented with 10% fetal bovine serum (FBS), 1% Penicillin-Streptomycin, and 1% L-Glutamine (200 mM). Cells were maintained in a humidified incubator with 5% CO_2_ at 37 °C.

### 2.3. Cell Seeding

For the seeding density optimization, cells were harvested during exponential growth, counted using an automated cell counter (Countess 3, Invitrogen, Bothell, WA, USA), and resuspended in appropriate culture media. To determine the optimal density for each cell line, cells were seeded in 100 µL into 96-well plates at various densities (125, 250, 500, 1000, 2000, 4000, and 8000 cells per well). A stock suspension of 1 × 10^6^ cells/mL was used to prepare 80,000 cells/mL by mixing 160 µL of the stock with 1.84 mL of culture medium (final volume: L). Serial dilutions were then performed to obtain the subsequent cell suspensions. To ensure proper control, each 96-well plate included wells with only medium without cells (blank control). Each condition was performed in triplicate. Cells were allowed to adhere and grow for 24, 48, and 72 h before cell viability analyses.

### 2.4. MTT Assay Procedure

Cell viability was assessed using an MTT assay in accordance with the manufacturer’s protocol. Briefly, at each designated time point (24, 48, and 72 h), 10 µL of MTT reagent was added to each well containing cells or blank control, and plates were then incubated for 4 h at 37 °C to ensure sufficient formazan formation, without saturation, based on established protocols [6,31]. Following incubation, formazan crystals were dissolved in 100 µL solubilization solution by gentle shaking on a gyratory shaker. Absorbance was measured at 570 nm with a reference wavelength of 630 nm using a microplate reader (Synergy H1, BioTek, Winooski, VT, USA).

### 2.5. Calibration Curve and Data Analysis

To select the optimal cell density, a standard curve was generated by seeding known cell densities (ranging from 125 to 8000 cells per well) and measuring their respective absorbance after the MTT assay at 24, 48, and 72 h. Linear regression analysis was performed to establish the linear correlation between the cell number and MTT-induced absorbance, allowing for the selection of an optimal cell number for cell viability experiments.

### 2.6. Assessment of Solvent Cytotoxicity

Stock solutions of DMSO and ethanol (100% concentration) were used to prepare working dilutions in culture media to achieve the following concentrations: 5%, 2.5%, 1.25%, 0.625%, and 0.3125% (*v*/*v*). For cytotoxicity analysis, cells were allowed to adhere and grow for 24 h, then the culture medium was replaced with an equal volume (100 µL) of DMSO or ethanol at different concentrations and allowed to grow for 24, 48, and 72 h before subsequent MTT assay analyses. Each 96-well plate included cells not treated with any solvents as a control group.

The cytotoxic effects of DMSO and ethanol at different concentrations were analyzed by comparing treated wells to untreated control wells. Data were expressed as percentage viability relative to the control. Cytotoxicity threshold was defined as more than 30% reduction in cell viability in accordance with ISO 10993-5:2009 [30].

### 2.7. Molecular Docking Studies

The ligands in the current study are DMSO and ethanol; the three-dimensional structures of the ligands were retrieved from the PubChem database [32]. For conducting the docking analysis, the receptors were selected under three categories: Drug Metabolizing Enzymes, and Transport Proteins (Cytochrome P450 2E1, PDB ID: 3E6I_A and ATP-Binding Cassette Subfamily B Member (ABCB1), P-glycoprotein (P-gp), PDB ID: 6QEX_A), Apoptosis Proteins (Caspase-3: PDB ID, 3KJF_A and BCL2 Associated X Protein (BAX), PDB ID: 1F16), and Membrane and Lipid Interaction Proteins (Phospholipase A2 (PLA2G4A), PDB ID: 1DB4_A and Liver X Receptor alpha (LXRα), Nuclear Receptor Subfamily 1 Group H Member 3 (NR1H3), PDB ID: 1UHL_A). All the receptor proteins were cleaned and prepared using Integrated Genetic Evolutionary Docking (iGemDock) v2.1 software. The binding site was selected as the site of the bound inhibitor in the proteins. Standard docking with a 200 population size and 70 generations was executed by employing the iGemDock v2.1 tool.

### 2.8. Intra-Molecular Interaction Studies

The ligands docked within the binding site of the receptors were further analyzed for intra-molecular interactions using Discovery Studio Visualizer v24 and Python Molecular Tool (PyMol) v. 2.5 [33].

### 2.9. Statistical Analysis

Data are presented as mean ± SD, and results were analyzed using one-way Analysis of Variance (ANOVA) to evaluate differences across multiple groups (e.g., cell lines, different solvent concentrations, and time points), followed by Tukey’s Post hoc comparisons test to identify specific differences between groups using GraphPad Prism v10.2.0. Statistical significance was considered at *p* < 0.05.

## 3. Results

### 3.1. Optimization of Cell Density for Accurate Viability Assessment

To establish the most suitable seeding density for reliable MTT assay measurements across the six cancer cell lines, HepG2, Huh7, HT29, SW480, MCF-7, and MDA-MB-231, cells were seeded at densities of 125, 250, 500, 1000, 2000, 4000, and 8000 cells per well. The assays were performed at 24, 48, and 72 h to assess the linearity of the relationship between cell number and absorbance. All cell lines demonstrated strong linear correlations (R^2^ = 0.95–0.99) between cell number and absorbance up to 4000 cells per well at each time point (Figure 1). This indicates that, within this range, the assay provides accurate quantification of cell viability. However, although linearity was maintained at higher densities, the signal at 4000 cells per well approached a plateau, suggesting potential saturation effects that could limit the assay’s sensitivity and accuracy at higher cell numbers. Lower densities, particularly 2000 cells per well, also exhibited excellent linearity within the linear portion of the plot, with R^2^ values comparable to those at 4000 cells, but offered a broader dynamic range without nearing saturation. Considering robust linearity, optimal signal-to-noise ratio, and reduced risk of signal saturation, a seeding density of 2000 cells per well was selected for subsequent experiments. This choice ensures accurate, reproducible, and sensitive measurement of cell viability across all cell lines and time points.

### 3.2. Effect of DMSO and Ethanol on Cell Viability in Various Cell Lines

The cytotoxic effects of DMSO and ethanol at various concentrations, 0.3125%, 0.625%, 1.25%, 2.5%, and 5% (*v*/*v*), were evaluated across six cancer cell lines (HepG2, Huh7, HT29, SW480, MCF-7, and MDA-MB-231) at 24, 48, and 72 h. One-way ANOVA revealed significant effects of both solvents on cell viability in all cell lines at all time points (*p* < 0.05). Cytotoxicity was defined as a reduction in cell viability exceeding 30% compared to untreated controls, in accordance with ISO 10993-5.

DMSO demonstrated a concentration- and time-dependent cytotoxicity profile (Table 1 and Figure 2). In HepG2 cells, DMSO showed cytotoxicity at the concentration of 2.5% at 24 and 48 h, with viability reductions exceeding 30% (41.6% ± 5.8, *p <* 0.0001; 42.8% ± 4.3, *p =* 0.0004, respectively; Figure 2, panels A1 and A2). At 72 h, cytotoxicity was observed at the concentration of 0.625% (33.6% ± 5.1, *p <* 0.0001), while the lowest concentration tested, 0.3125%, did not induce viability reduction at any time point (Figure 2, panel A3). Huh7 cells exhibited cytotoxicity at 5% DMSO at 24 (49.1% ± 0.35, *p <* 0.0001), (62.7 % ± 5.4, *p =* 0.0001, respectively), with 2.5% and 5% causing cytotoxic effects at 48 h (31.7 ± 3.9, *p =* 0.0001; 62.7 % ± 5.4, *p =* 0.0001, respectively) and 72 h (46.6% ± 1.09, *p <* 0.0001; 93.29% ± 0.3, *p <* 0.0001, respectively; Figure 2, panels B1–B3). The lower concentrations of ≤1.25% did not produce a >30% reduction in viability at any time point. In HT29 cells, cytotoxicity was evident at 1.25% after 24 h (42.3% ± 4.2, *p <* 0.0001) and at 0.625% and higher at 48 and 72 h (33.9% ± 0.6, *p <* 0.0001; 38% ± 1.7, *p <* 0.0037, respectively; Figure 2, panels C1–C3). SW480 cells showed >30% viability reductions at 5% after 24 h (59.5% ± 4.2, *p <* 0.0001) and at 2.5% and above at 48 and 72 h (39% ± 1.6, *p <* 0.0001; 36.8% ± 12.9, *p =* 0.0005, respectively; Figure 2, panels D1–D3). In MCF-7 cells, 3125% and 0.625% DMSO were not cytotoxic at 24 h (11.9% ± 2.4, *p =* 0.01; 20.2% ± 2, *p =* 0.0003, respectively). However, at 48 and 72 h, all concentrations (except 0.625% at 48 h, 27.4% ± 2.4, *p <* 0.0001) were cytotoxic (Figure 2, panels E1–E3). MDA-MB-231 cells were relatively resistant, with only 5% DMSO showing a cytotoxic effect at 24 h (53% ± 1.4, *p <* 0.0001). In contrast, 2.5% and 5% ethanol caused cytotoxicity at 48 h (62.7% ± 7.5, *p <* 0.0001; 87.8% ± 2.9, *p <* 0.0001, respectively) and 72 h (43% ± 8.4, *p <* 0.0001; 97.1% ± 0.61, *p <* 0.0001, respectively; Figure 2, panels F1–F3).

Ethanol had a more pronounced cytotoxic effect across all cell lines (Table 2 and Figure 3). In HepG2 cells, cytotoxicity was observed at 1.25% and above at 24 and 48 h (48.3.7% ± 8, *p <* 0.0001; 38.6% ± 5, *p <* 0.0001, respectively), with all tested concentrations causing a viability reduction greater than 30% at 72 h (Figure 3, panels A1–A3). Huh7 cells were highly sensitive, with all concentrations causing cytotoxicity at 24 h; at 48 h, 1.25% (38.5% ± 4.7, *p <* 0.0001) and above were cytotoxic, while, at 72 h, all concentrations were cytotoxic (Figure 3, panels B1–B3). HT29 cells showed cytotoxicity at all concentrations at 24 h. However, at 48 h, 2.5% and 5% were cytotoxic (49.3% ± 6.2, *p <* 0.0001; 80.3% ± 11.7, *p <* 0.0001, respectively), while, at 72 h, 1.25% (38.3% ± 3.8, *p =* 0.0002) and above induced cytotoxicity (Figure 3, panels C1–C3). SW480 cells experienced cytotoxic effects at all concentrations at both 24 and 48 h, but only 1.25% (35.3% ± 6.3, *p <* 0.0001) and above were cytotoxic at 72 h (Figure 3, panels D1–D3). Both MCF-7 and MDA-MB-231 cell lines exhibited a similar cytotoxic profile for ethanol. At 24 h, 0.3125% (MCF-7: 21% ± 7.3, *p =* 0.0004; MDA.MB.231: 9.8% ± 8.4, *p =* 0.54) and 0.625% (MCF-7: 24.2% ± 5.5, *p <* 0.0001; MDA.MB.231: 6.8% ± 1.4, *p =* 0.82) ethanol concentrations did not show cytotoxic effect. However, all concentrations were cytotoxic at 48 h, and only 0.3125% (MCF-7: 27.4% ± 2.8, *p <* 0.0001; MDA.MB.231: 14% ± 5.4, *p =* 0.01) did not exhibit cytotoxic effect at 72 h (Figure 3, panels E1–E3, F1–F3).

### 3.3. In Silico Analysis

The docking analysis of DMSO and ethanol with the drug metabolizing protein (PDB ID: 3E6I) revealed distinct differences in binding affinity and interaction profiles. Ethanol showed a higher binding affinity and stronger van der Waals interactions than DMSO with the drug-metabolizing protein. This study reveals that DMSO forms two conventional hydrogen bonds with LEU463 and VAL464, while ethanol forms multiple hydrogen and hydrophobic bonds with ARG100, ARG435, ILE114, and ILE115. This combination of polar and non-polar interactions enhances binding specificity and stability, resulting in lower docking energy. Additionally, this study found that DMSO and ethanol exhibited a better interaction energy with another drug metabolizing protein (PDB ID: 6QEX) than 3E6I. This study reveals that DMSO forms five conventional hydrogen bonds with residues LYS1076, SER1077, and THR1078, favoring interaction with the sulfoxide group. Ethanol forms five conventional hydrogen bonds with residues GLY430, CYS431, GLY432, LYS433, and ASN428, all polar (Table 3 and Figure 4).

Additionally, this study explored the interaction of DMSO and ethanol with the membrane protein (PDB ID: 1DB4). The analysis revealed that DMSO had a slightly higher binding affinity and stronger van der Waals interactions compared to ethanol for the 1DB4. DMSO formed five key hydrogen bonds with THR61, LYS62, and PHE63 atoms, while ethanol formed five hydrogen bonds with CYS50, CYS90, and CYS90 atoms. DMSO’s better binding energy may be attributed to its dominance in polar hydrogen bonding, whereas ethanol exhibits a weaker van der Waals component due to its hydrophobic alkyl interactions. Furthermore, the interaction between DMSO and ethanol with another membrane protein (PDB ID: 1UHL) was studied and found a higher binding affinity and stronger van der Waals interactions of DMSO with 1UHL. DMSO formed three conventional hydrogen bonds with SER366 and ARG415, while ethanol formed six hydrogen bonds with ARG415, GLU296, and LEU411. DMSO’s lower docking energy was attributed to its shorter and more specific polar contacts, suggesting a better fit and stronger stabilization in the binding site (Table 4 and Figure 5).

The intra-molecular interaction of DMSO and ethanol with the apoptotic proteins was evaluated. The apoptotic protein (PDB ID: 3KJF) exhibited slightly stronger binding affinity with DMSO compared to ethanol. Additionally, DMSO showed a lower van der Waals (vDW) than ethanol. In the 3KJF–DMSO complex, three conventional hydrogen bonds were formed involving residues LYS260 and ASP169. Moreover, pi-alkyl interactions were observed between the ligand and aromatic residues TYR203 and TRP206, contributing to hydrophobic stabilization. In contrast, the 3KJF–ethanol complex displayed two conventional hydrogen bonds, formed with ARG64 and ARG207, respectively. While still energetically favorable, the reduced number and diversity of interactions suggest slightly weaker binding compared to DMSO. Additionally, for another apoptotic protein (PDB ID: 1F16), DMSO again demonstrated a stronger binding affinity and favorable van der Waals energy than ethanol. The 1F16–DMSO complex formed two strong conventional hydrogen bonds with ARG109, with exceptionally short distances of 2.19 Å and 2.25 Å, indicating strong polar interactions. Additional stabilizing forces included pi-sigma and pi-sulfur interactions with TRP151, reflecting aromatic and sulfur-mediated stabilization mechanisms. The 1F16–ethanol complex, although slightly weaker in energy, exhibited three conventional hydrogen bonds with ASN73 and LEU70, in addition to carbon-hydrogen, alkyl, and pi-alkyl interactions with residues such as ASP71, LEU76, ILE80, and TYR115 (Table 5 and Figure 6).

## 4. Discussion

This study underscores the importance of meticulous methodological optimization, particularly regarding cell density and solvent concentrations, to enhance the reproducibility and reliability in drug efficacy evaluation. To the best of our knowledge, this is the first comprehensive study investigating the cytotoxic effects of DMSO and ethanol on this set of cancer cell lines, covering hepatocellular carcinoma (HCC), breast cancer, colorectal cancer (CRC), at multiple time points. The study design was structured to systematically assess the effects of solvents (DMSO and ethanol) across multiple cell lines and time points, ensuring comprehensive and applicable findings. We selected six cancer cell lines (HepG2, Huh7, HT29, SW480, MCF-7, and MDA-MB-231) due to their widespread use in cancer research and their distinct biological characteristics [34,35,36,37,38,39]. These cell lines represent HCC, CRC, and breast cancer, providing a diverse model set to evaluate solvent effects across different tumor types. The selection was driven by the need to understand solvent cytotoxicity in varied cellular contexts, facilitating broader applicability for our findings. We evaluated the effects of DMSO and ethanol on cell viability across these cell lines, aiming to identify optimal solvent concentrations for in vitro experiments, complemented by in silico docking analyses to provide mechanistic insights into solvent–protein interactions. A key aspect of this investigation was the preliminary optimization of cell seeding densities, which revealed that a density of 2000 cells per well provided a reliable and reproducible basis for assessing cytotoxicity (Figure 1). This density produced consistent and linear viability across all cell lines and time points, aligning with established protocols that emphasize the importance of optimal cell density for assay accuracy [6,40,41]. Maintaining appropriate cell density is critical to minimize variability, prevent nutrient depletion, and avoid saturation effects that could confound toxicity assessments [6,7].

The subsequent in vitro cytotoxicity assessment demonstrated that both DMSO and ethanol exert significant concentration and time-dependent cytotoxicity on different types of cancer cell lines (Figure 2 and Figure 3). DMSO exhibited a gradual, concentration- and time-dependent cytotoxic profile on the tested cell lines, which is consistent with its known ability to modulate apoptosis pathways [42,43]. It has been shown previously that DMSO at concentrations of ≤1% exerts minimal impact on cell health [14,23,44,45]. For example, Moskot et al. reported that 1% DMSO concentration showed a non-cytotoxic effect on human fibroblasts [14]. Similarly, Sangweni et al. demonstrated that DMSO concentration ≤0.5% had no significant cytotoxic effect on cardiac and cancer cells [45]. Our findings reveal that DMSO concentration 0.3125% was non-cytotoxic (<30% cell viability reduction) across all cell lines and time points, consistent with previous reports [14,23,45]. However, the MCF-7 cell line displayed increased sensitivity to the lowest DMSO concentration (0.3125%) at 48 and 72 h, leading to a cell viability reduction exceeding 30%. The displayed cytotoxicity observed at lower DMSO concentrations after 48 and/or 72 h in all cell lines may be attributed to the gradual induction of oxidative stress and apoptosis by DMSO. Previous studies have demonstrated that DMSO can modulate mitochondrial function, reactive oxygen species (ROS), and activate apoptosis pathways [42,45]. Moreover, differences in cell membrane composition and metabolic rate may underlie the variable sensitivity observed among cell lines. For instance, HepG2 and HT29 cells, known for their high metabolic activity, may accumulate DMSO’s effects more rapidly, leading to increased cytotoxicity over time [46,47,48]. In contrast, MDA-MB-231 and Huh7 cells exhibited greater resistance, possibly due to more robust antioxidant defenses or differences in solvent uptake and efflux mechanisms [49,50,51,52,53].

Ethanol demonstrated a more rapid and potent effect across all six cancer cell lines, inducing a significant reduction in cell viability (>30%) at lower concentrations and earlier time points compared to DMSO (Table 2). Specifically, ethanol exerted a cytotoxic effect, causing substantial cell death at concentrations as low as 0.3125% at 24 h post-treatment in three cell lines (Huh7, HT29, and SW480), with viability dropping below 70%. These results are consistent with Worley et al.’s study, indicating that low concentrations of ethanol can induce over 30% cell death after 24 h in various cell lines [54]. The findings herein suggest that ethanol’s cytotoxic mechanism may predominantly involve membrane disruption and metabolic interference, leading to rapid cell death [21,55]. Furthermore, Truong et al. reported that ethanol exhibited non-toxic effects on HepG2, MCF-7, and MDA-MB-231 cell lines at concentrations ranging from 1.25% to 0.15% [23]. Our findings are consistent with this, showing that ethanol concentrations of 0.625% and 0.3125% are non-cytotoxic at 24 h on these cell lines. Interestingly, all cell lines (except HepG2) demonstrated reduced cytotoxicity at later time points when exposed to lower concentrations of ethanol (Figure 3). This observation may reflect the cellular adaptation mechanisms or selective survival of resistant subpopulations [56,57]. In contrast, HepG2 cells did not exhibit this trend, which may be explained by differences in metabolic capacities or stress response pathways [58].

The in vitro study was further investigated by an in silico approach. The in silico study explored intra-molecular interactions of DMSO and ethanol with critical proteins associated with drug metabolism and transportation, apoptosis, and membrane and lipid interaction proteins from HCC, CRC, and breast cancer. DMSO and ethanol are often considered inert in experimental settings [18,59,60]. However, our results indicate that they can interact significantly with key cancer-related proteins, potentially affecting cellular behavior and therapeutic responses. This highlights the importance of understanding solvent–protein interactions in cancer research, as they may influence the outcomes of in vitro studies. The in silico docking analyses supported in vitro observations, revealing that DMSO binds strongly and specifically to key apoptotic and membrane proteins, forming multiple hydrogen bonds and hydrophobic interactions with residues critical for protein function (Figure 5 and Figure 6). This indicates that DMSO may induce apoptosis through direct modulation of these proteins, which aligns with its slower but targeted cytotoxic effect observed in vitro. Ethanol, while exhibiting higher overall binding affinity to drug-metabolizing enzymes, showed weaker and less specific interactions with apoptotic proteins, which correlates with its rapid membrane-disruptive cytotoxicity rather than apoptosis induction [21,61]. Overall, the combined in vitro and in silico data suggest that ethanol exerts immediate cytotoxic effects primarily via membrane destabilization. In contrast, DMSO induces apoptosis through targeted protein interactions, leading to a delayed cytotoxic response. This integrated understanding underscores the importance of considering both direct membrane effects and protein-mediated pathways in evaluating solvent cytotoxicity.

The strengths of this study lie in its comprehensive and quantitative assessment of the cytotoxic thresholds of DMSO and ethanol in different cancer cell lines over multiple time points, providing practical guidelines for solvent use in in vitro assays. Our integration of experimental viability data with in silico mechanistic insights offers a deeper understanding of the molecular interactions underlying solvent-induced cytotoxicity, emphasizing that solvents such as DMSO and ethanol may influence experimental outcomes. However, limitations include the use of only a limited set of cell lines, which may not represent the full heterogeneity of tumor responses. Additionally, while an in silico model is informative, it requires validation through molecular techniques to assess other cellular responses, such as apoptosis and oxidative stress.

## 5. Conclusions

This study demonstrated that optimizing cell density and solvent concentrations at different periods is crucial for obtaining reproducible and reliable in vitro cytotoxicity data. Our findings revealed that DMSO at 0.3125% is suitable for minimizing cytotoxic effects in all cell lines and time points. However, MCF-7 cells demonstrated high sensitivity to this concentration at 48 and 72 h, indicating that the non-cytotoxic threshold of DMSO concentration is dependent on cell type and exposure duration. Moreover, ethanol exhibited higher toxicity at even lower concentrations, with significant reductions in cell viability observed at 0.3125% as early as 24 h in several cell lines, highlighting the need for careful concentration management due to its rapid and broad cytotoxic effects. Additionally, the in silico analyses supported these observations by revealing that ethanol’s stronger and more diverse interactions with key proteins involved in apoptosis and metabolism likely underlie its higher cytotoxicity. Importantly, this integrated approach highlights that optimizing experimental parameters, such as cell density and solvent concentration, not only enhances reproducibility but also ensures a more accurate interpretation of results, thereby reducing variability and confounding factors in cell-based assays.

## Figures and Tables

**Figure 1 mps-08-00093-f001:**
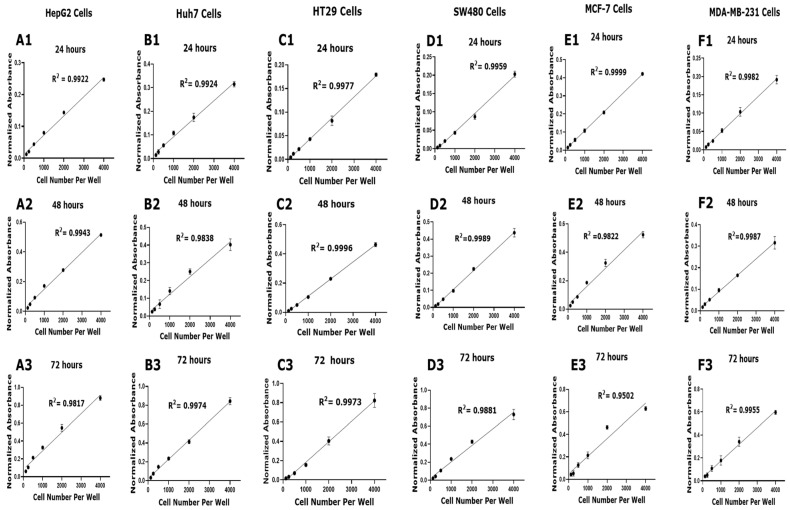
Cell viability standard curves for each cell line. Cell viability of each cell line was assessed using the MTT assay at various time points: panels (**A1**–**F1**) at 24 h; panels (**A2**–**F2**) at 48 h; panels (**A3**–**F3**) at 72 h. Linear regression was performed to assess the relationship between cell number density and the absorbance at each time point. Data presented are mean ± SD (*n* = 3). R-squared correlation coefficient (R^2^) is shown for each curve, indicating the linearity and reliability of the assay for quantifying viable cells across tested densities.

**Figure 2 mps-08-00093-f002:**
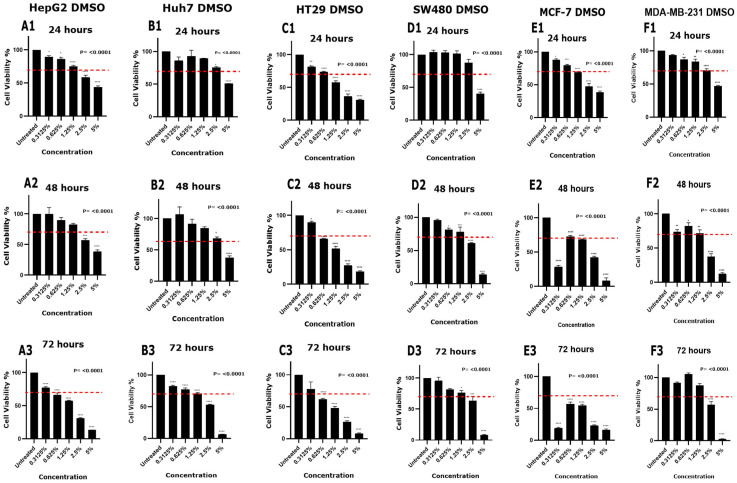
Cytotoxic effect of DMSO on each cell line. Cell viability of each cell line was assessed using an MTT assay at various time points [(panels (**A1**–**F1**) at 24 h), (panels (**A2**–**F2**) at 48 h), and (panels (**A3**–**F3**) at 72 h)]. Bar graphs represent cell viability for each condition (cell line, DMSO concentrations, and time points. The horizontal dashed red line indicates the 70% viability threshold. Data presented are mean ± SD (*n* = 3). Significance levels derived from one-way ANOVA (*p* < 0.0001) on graphs and annotations above bars derived from Tukey’s Post hoc comparisons (* *p <* 0.05, ** *p <* 0.01, *** *p <* 0.001, **** *p <* 0.0001).

**Figure 3 mps-08-00093-f003:**
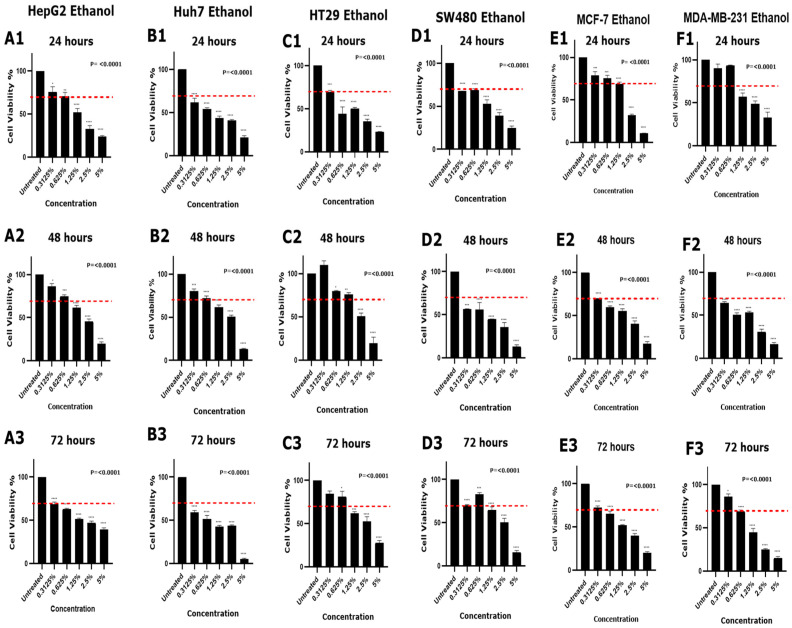
Cytotoxic effect of ethanol on each cell line. Cell viability of each cell line was assessed using an MTT assay at various time points [(panels (**A1**–**F1**) at 24 h), (panels (**A2**–**F2**) at 48 h), and (panels (**A3**–**F3**) at 72 h)]. Bar graphs represent cell viability for each condition (cell line, ethanol concentrations, and time points. The horizontal dashed red line indicates the 70% viability threshold. Data presented are mean ± SD (*n* = 3). Significance levels derived from one-way ANOVA (*p* < 0.0001) on graphs and annotations above bars derived from Tukey’s Post hoc comparisons (* *p <* 0.05, ** *p <* 0.01, *** *p <* 0.001, **** *p <* 0.0001).

**Figure 4 mps-08-00093-f004:**
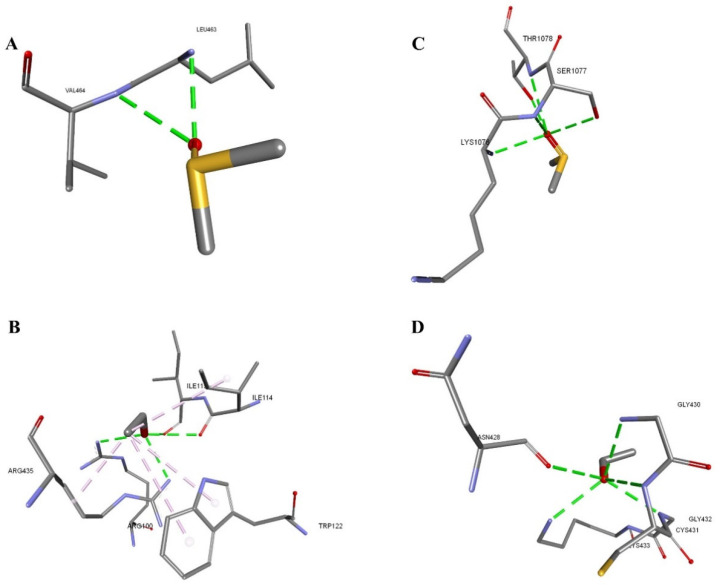
DMSO and ethanol docked in the drug-metabolizing proteins. Interacting residues are shown in sticks, and bonds are shown in dashed lines. (**A**) 3E6I-DMSO, (**B**) 3E6I-Ethanol, (**C**) 6QEX-DMSO, (**D**) 6QEX-Ethanol.

**Figure 5 mps-08-00093-f005:**
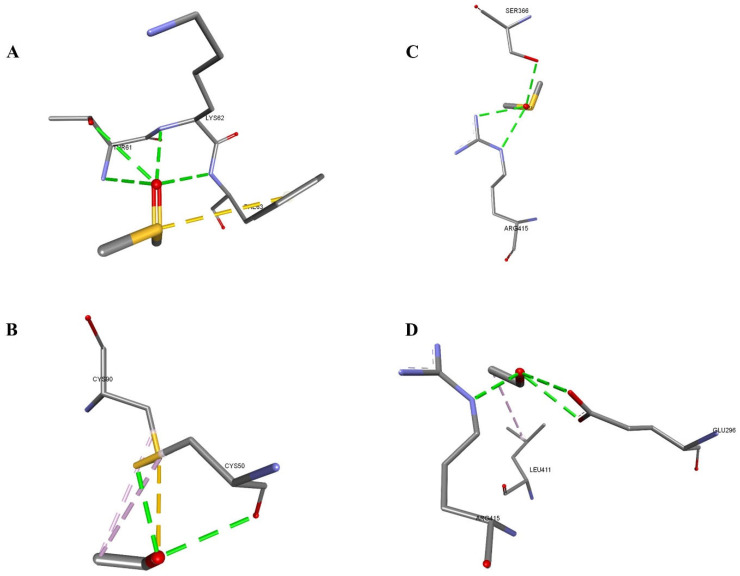
DMSO and ethanol docked in the drug membrane proteins. Interacting residues are shown in sticks, and bonds are shown in dashed lines. (**A**) 1DB4-DMSO, (**B**) 1DB4-Ethanol, (**C**) 1UHL-DMSO, (**D**) 1UHL-Ethanol.

**Figure 6 mps-08-00093-f006:**
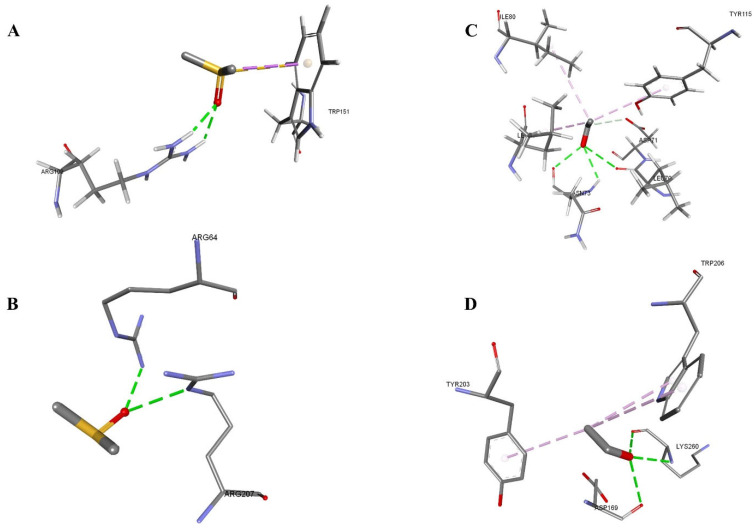
DMSO and ethanol docked in the drug apoptosis proteins. Interacting residues are shown in sticks, and bonds are shown in dashed lines. (**A**) 1F16-DMSO, (**B**) 1F16-Ethanol, (**C**) 3KGF-DMSO, (**D**) 3KGF-Ethanol.

**Table 1 mps-08-00093-t001:** Summary of DMSO cytotoxicity in cancer cell lines at multiple time points.

DMSO Concentration	Cytotoxicity Effect (>30% Reduction in Cell Viability) *																	
	HepG2			Huh7			HT29			SW480			MCF-7			MDA-MB-231		
	24 h	48 h	72 h	24 h	48 h	72 h	24 h	48 h	72 h	24 h	48 h	72 h	24 h	48 h	72 h	24 h	48 h	72 h
0.3125%	No	No	No	No	No	No	No	No	No	No	No	No	No	Yes	Yes	No	No	No
0.625%	No	No	Yes	No	No	No	No	Yes	Yes	No	No	No	No	No	Yes	No	No	No
1.25%	No	No	Yes	No	No	No	Yes	Yes	Yes	No	No	No	Yes	Yes	Yes	No	No	No
2.5%	Yes	Yes	Yes	No	No	Yes	Yes	Yes	Yes	No	Yes	Yes	Yes	Yes	Yes	No	Yes	Yes
5%	Yes	Yes	Yes	Yes	Yes	Yes	Yes	Yes	Yes	Yes	Yes	Yes	Yes	Yes	Yes	Yes	Yes	Yes

* No (green) = non-cytotoxic; Yes (red) = cytotoxic.

**Table 2 mps-08-00093-t002:** Summary of ethanol cytotoxicity in cancer cell lines at multiple time points.

Ethanol Concentration	Cytotoxicity Effect (>30% Reduction in Cell Viability) *																	
	HepG2			Huh7			HT29			SW480			MCF-7			MDA-MB-231		
	24 h	48 h	72 h	24 h	48 h	72 h	24 h	48 h	72 h	24 h	48 h	72 h	24 h	48 h	72 h	24 h	48 h	72 h
0.3125%	No	No	Yes	Yes	No	Yes	Yes	No	No	Yes	Yes	No	No	Yes	No	No	Yes	No
0.625%	No	No	Yes	Yes	No	Yes	Yes	No	No	Yes	Yes	No	No	Yes	Yes	No	Yes	Yes
1.25%	Yes	Yes	Yes	Yes	Yes	Yes	Yes	No	Yes	Yes	Yes	Yes	Yes	Yes	Yes	Yes	Yes	Yes
2.5%	Yes	Yes	Yes	Yes	Yes	Yes	Yes	Yes	Yes	Yes	Yes	Yes	Yes	Yes	Yes	Yes	Yes	Yes
5%	Yes	Yes	Yes	Yes	Yes	Yes	Yes	Yes	Yes	Yes	Yes	Yes	Yes	Yes	Yes	Yes	Yes	Yes

* No (green) = non-cytotoxic; Yes (red) = cytotoxic.

**Table 3 mps-08-00093-t003:** Intramolecular interactions between drug-metabolizing proteins and the solvents.

Protein–Solvent	Energy (kcal/mol)	vdW (kcal/mol)	Interacting Residues (Protein-Ligand)	Distance (Å)	Bond Type
**3E6I–DMSO**	−32.443	−25.443	A:LEU463:N–Z:PRE999:O	2.65103	Conventional H-Bond
			A:VAL464:N–Z:PRE999:O	3.04291	Conventional H-Bond
**3E6I–Ethanol**	−37.3332	−20.248	A:ARG100:NH1–Z:PRE999:O	2.61927	Conventional H-Bond
			A:ARG435:NH1–Z:PRE999:O	2.65332	Conventional H-Bond
			Z:PRE999:O–A:ILE114:O	2.86426	Conventional H-Bond
			Z:PRE999:O–A:ILE115:O	3.13713	Conventional H-Bond
			Z:PRE999:C–A:ILE114	4.96809	Alkyl
			Z:PRE999:C–A:ARG435	3.64147	Alkyl
			A:TRP122–Z:PRE999:C	4.86274	Pi-Alkyl
			A:TRP122–Z:PRE999:C	5.38205	Pi-Alkyl
**6QEX–DMSO**	−36.256	−20.7804	A:LYS1076:N–Z:PRE999:O	2.84994	Conventional H-Bond
			A:SER1077:N–Z:PRE999:O	2.8904	Conventional H-Bond
			A:SER1077:OG–Z:PRE999:O	3.10488	Conventional H-Bond
			A:THR1078:N–Z:PRE999:O	2.94488	Conventional H-Bond
			A:THR1078:OG1–Z:PRE999:O	2.70126	Conventional H-Bond
**6QEX–Ethanol**	−35.8352	−18.9044	A:GLY430:N–Z:PRE999:O	3.08366	Conventional H-Bond
			A:CYS431:N–Z:PRE999:O	2.9703	Conventional H-Bond
			A:GLY432:N–Z:PRE999:O	2.78409	Conventional H-Bond
			A:LYS433:NZ–Z:PRE999:O	2.89953	Conventional H-Bond
			Z:PRE999:O–A:ASN428:O	3.10461	Conventional H-Bond

**Table 4 mps-08-00093-t004:** Intramolecular interactions between drug membrane proteins and the solvents.

Protein–Solvent	Energy	vDW	Interacting Residues (Protein-Ligand)	Distance (Å)	Bond Type
1DB4-DMSO	−33.14	−20.61	A:THR61:N-Z:PRE999:O	3.05995	Conventional Hydrogen Bond
			A:THR61:OG1-Z:PRE999:O	3.19181	Conventional Hydrogen Bond
			A:LYS62:N-Z:PRE999:O	2.62457	Conventional Hydrogen Bond
			A:PHE63:N-Z:PRE999:O	3.10118	Conventional Hydrogen Bond
			Z:PRE999:S-A:PHE63	4.61255	Pi-Sulfur
1DB4-Ethanol	−30.52	−17.0	Z:PRE999:O-A:CYS50:O	3.09723	Conventional Hydrogen Bond
			Z:PRE999:O-A:CYS50:SG	2.71761	Conventional Hydrogen Bond
			A:CYS90:SG-Z:PRE999:O	3.24338	Sulfur-X
			Z:PRE999:C-A:CYS50	4.47882	Alkyl
			Z:PRE999:C-A:CYS90	4.41033	Alkyl
1UHL-DMSO	−37.78	−21.95	B:SER366:OG-Z:PRE999:O	2.88224	Conventional Hydrogen Bond
			B:ARG415:NE-Z:PRE999:O	2.97604	Conventional Hydrogen Bond
			B:ARG415:NH2-Z:PRE999:O	2.82177	Conventional Hydrogen Bond
1UHL-Ethanol	−36.06	−15.27	B:ARG415:NE-Z:PRE999:O	2.71453	Conventional Hydrogen Bond
			Z:PRE999:O-B:GLU296:OE1	3.11114	Conventional Hydrogen Bond
			Z:PRE999:O-B:GLU296:OE2	3.1035	Conventional Hydrogen Bond
			Z:PRE999:C-B:LEU411	4.09698	Alkyl

**Table 5 mps-08-00093-t005:** Intramolecular interactions between drug apoptosis proteins and the solvents.

Protein–Solvent	Energy	vDW	Interacting Residues (Protein-Ligand)	Distance (Å)	Bond Type
3KJF-DMSO	−38.13	−26.53	B:LYS260:N-Z:PRE999:O	3.13923	Conventional Hydrogen Bond
			Z:PRE999:O-A:ASP169:O	3.12752	Conventional Hydrogen Bond
			Z:PRE999:O-B:LYS260:O	2.59778	Conventional Hydrogen Bond
			B:TYR203-Z:PRE999:C	4.93447	Pi-Alkyl
			B:TRP206-Z:PRE999:C	4.60801	Pi-Alkyl
			B:TRP206-Z:PRE999:C	5.45039	Pi-Alkyl
3KJF-Ethanol	−37.01	−21.94	A:ARG64:NH2-Z:PRE999:O	2.60446	Conventional Hydrogen Bond
			B:ARG207:NE-Z:PRE999:O	3.1156	Conventional Hydrogen Bond
1F16-DMSO	−34.49	−27.49	A:ARG109:HH12-Z:PRE999:O	2.19023	Conventional Hydrogen Bond
			A:ARG109:HH22-Z:PRE999:O	2.25436	Conventional Hydrogen Bond
			Z:PRE999:C-A:TRP151	3.66643	Pi-Sigma
			Z:PRE999:S-A:TRP151	3.84053	Pi-Sulfur
1F16-Ethanol	−33.12	−22.77	A:ASN73:H-Z:PRE999:O	2.66065	Conventional Hydrogen Bond
			Z:PRE999:O-A:LEU70:O	3.0568	Conventional Hydrogen Bond
			Z:PRE999:O-A:ASN73:O	2.52568	Conventional Hydrogen Bond
			Z:PRE999:C-A:ASP71:OD1	3.70876	Carbon Hydrogen Bond
			Z:PRE999:C-A:LEU76	3.84647	Alkyl
			Z:PRE999:C-A:ILE80	5.19701	Alkyl
			A:TYR115-Z:PRE999:C	5.24314	Pi-Alkyl

## Data Availability

Data presented in this study are available in the article.

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
