# Peer review of "Optimizing Cell Density and Unveiling Cytotoxic Profiles of DMSO and Ethanol in Six Cancer Cell Lines: Experimental and In Silico Insights"

_mps, 2025, doi:10.3390/mps8040093_

Round 1
Reviewer 1 Report
Comments and Suggestions for Authors
The manuscript is well argued, emphasizing the need to identify optimal densities for culturing different cell lines, in this study, six tumor lines. The authors also evaluated the cytotoxicity of the most commonly used solvents, DMSO and ethanol, representing a important setup for in vitro tests.
The work is well written and can be read fluently, however a few comments are inserted in PDF manuscript.

Author Response
Thank you for your valuable comments. All comments in the PDF have been addressed as advised.
Reviewer 2 Report
Comments and Suggestions for Authors
Comments
The work of Asiri A, et al. titled “Optimizing Cell Density and Unveiling Cytotoxic Profiles of DMSO and Ethanol in Six Cancer Cell Lines: Experimental and In Silico Insights” describes a systematic approach to evaluate the effect of cell density and two different solvents on the reproducibility of cytotoxicity studies. It is interesting work since most researchers do not take into account the importance of cytotoxicity assays optimization depending on the different cell lines and compounds under study. Thus, it opens our mind to the importance of such optimizations and guides us some guidelines to perform these studies. It fits perfectly into the scope of Methods and Protocols.
Major comments
- Did the authors consider also performing in vitro assays related to apoptosis, for instance, to corroborate the in silico data?
Minor comments
Introduction
- It would be interesting to describe a little bit more MTT and give more information on the reason why it was selected among all the other methodologies.
- “DMSO and ethanol as carriers.” – I think that carriers is not the best work to describe these solvents.
- Do the authors think or consider the incubation time with MTT?
Methods
- Section 2.1 – Add the ATCC code of each cell line
Results
- Figure 1 must be redesigned for better understanding. Consider adding a general cell line name and incubation time in the same column or line. Use the same scale bars to better compare the results. Do you consider that the use of cells after 24h or 72h, for instance, is similar?
- Section 3.2 – Add concentration (V/V). Are you aware that the percentage of ethanol in the assays might not be what you say due to evaporation? Do you have any data indicating that after 72h, the ethanol concentration is similar to the stock?
- Figure 2 and 3 are too confusing… Consider the comments on figure 1.
Discussion
- very long paragraph, which makes the understanding more difficult.
Author Response
Major comments
- Did the authors consider also performing in vitro assays related to apoptosis, for instance, to corroborate the in silico data?
Thank you for highlighting this important point. Incorporating in vitro assays related to apoptosis would indeed provide valuable experimental validation of the in silico interactions and the mechanisms underlying solvent-induced cytotoxicity. We did not perform these additional assays due to the scope of our current study, which primarily focuses on determining non-cytotoxic concentrations of these solvents in different cancer cell lines. However, we acknowledge the importance of these assays and believe future work should include such analyses to further elucidate the apoptotic pathways involved. This point has been addressed as a limitation of the study at the end of the discussion section.
Minor comments
Introduction
- It would be interesting to describe a little bit more MTT and give more information on the reason why it was selected among all the other methodologies.
Thank you for your suggestion. In our revised introduction, we have included a brief description of the MTT assay, emphasizing its widespread use, reliability, simplicity, and quantitative nature for assessing cell viability. We have also clarified that it was chosen due to its cost-effectiveness, ease of execution, and established validation in numerous cytotoxicity studies, making it suitable for evaluating the impact of solvents on cell viability.
- “DMSO and ethanol as carriers.” – I think that carriers is not the best work to describe these solvents.
We agree that “carriers” may not be the most accurate term to describe these solvents. In the revised version, we have replaced “carriers” with “vehicles”, which more accurately describe their role in dissolving compounds for in vitro testing
- Do the authors think or consider the incubation time with MTT?
Thank you for your valuable comment. In our study, we used an incubation period of 4 hours with MTT reagent, which is standard in many protocols to allow sufficient conversion of MTT to formazan without saturation effects. We have clarified this in the methods section and briefly discussed the rationale for this incubation time, emphasizing its validation in previous studies for capturing mitochondrial activity reflective of cell viability.
Methods
- Section 2.1 – Add the ATCC code of each cell line
Thank you for pointing that out. The revised methods section now includes the ATCC code for each cell line.
Results
- Figure 1 must be redesigned for better understanding. Consider adding a general cell line name and incubation time in the same column or line. Use the same scale bars to better compare the results. Do you consider that the use of cells after 24h or 72h, for instance, is similar?
Thank you for this valuable suggestion. In the revised results, figures 1-3 have been redesigned as advised to improve clarity. Regarding scale bars, due to variations in OD values across different cell lines and time points, using the same scale bars would misrepresent the data and affect the visual interpretation of linearity in some plots. Therefore, they were adjusted accordingly to accurately reflect the specific OD ranges for each condition.
- Section 3.2 – Add concentration (V/V).
Thank you for pointing this out. (v/v) has now been added to section 3.2 in the revised results.
Are you aware that the percentage of ethanol in the assays might not be what you say due to evaporation? Do you have any data indicating that after 72h, the ethanol concentration is similar to the stock?
Thank you for highlighting this important point. In this study, ethanol solutions were freshly prepared for each experiment, and the cultures were maintained in sealed plates under standard conditions (37ºC, 5% CO2) to minimize evaporation. We did not specifically measure ethanol concentration in the culture medium after incubation. Although evaporation could potentially reduce ethanol concentrations over time, the significant cytotoxic effects observed at the tested concentrations suggest that ethanol activity remained sufficient to exert its biological effects throughout the exposure periods across different cell lines.
- Figure 2 and 3 are too confusing… Consider the comments on figure 1.
As mentioned in response to comments on the figure, the revised results show that figures 1-3 have been redesigned as advised for better clarity.
Discussion
- very long paragraph, which makes the understanding more difficult.
Thank you for your feedback. We recognize that the paragraph is lengthy and may hinder clarity. In the revised manuscript, we have divided the discussion into smaller, focused paragraphs to improve readability and understanding. This restructuring aims to clearly present our findings, comparisons with previous studies, and interpretations to enhance overall clarity without losing the comprehensive nature of the discussion.
Reviewer 3 Report
Comments and Suggestions for Authors
The line numbers are missing, so all comments should be made using an approximate localisation.
1. Abstract:
Please organize it in Objective, Materials and Methods, Results, and Conclusions. Please ensure that both solvents are mentioned in all sections. The keywords should include cancer cell lines; please remove 'cancer' and 'cell lines'. MTT assay is the complete name of the test; please correct.
2. Introduction:
The authors stated: "In drug discovery studies, solvents like Dimethyl Sulfoxide (DMSO) and ethanol are commonly used as carriers or controls [12]. However, these solvents possess intrinsic cytotoxic properties that can confound experimental outcomes by amplifying or mimicking drug effects [13,14]. Moreover, different cell lines exhibit variable sensitivities to these solvents, which can potentially lead to misleading interpretations [15-18].
However, the authors should mention that most in vitro experiments use ethanol and DMSO in the cell medium at a concentration < 0.1% and include these solvents as controls to ensure accuracy.
They are invited to find and provide suitable references, as the cytotoxicity of these solvents is well-documented: https://scholar.google.com/scholar?hl=ro&as_sdt=0%2C5&q=dmso+cytotoxicity+on+cells&oq=dmso+cyt
https://scholar.google.com/scholar?hl=ro&as_sdt=0%2C5&q=ethanol+cytotoxicity+on+cells&btnG=
Therefore, the authors are invited to conduct a literature review in the Introduction, providing evidence of the current findings on the cytotoxicity of DMSO and ethanol, as well as the underlying mechanisms involved.
Then, they could display the hypotheses for their study and highlight its novelty and valuable contribution to the scientific database.
3. Materials and Methods
The authors are invited to maintain the original title of this section, because they used both materials and methods, not only methods.
The first subsection should be titled "Materials," where it should include all cell lines and their provenance (complete company name, city, state, and country), reagents, standards, kits for analysis, culture media, consumables, etc.
Then, they should describe the methods used, with suitable references.
4. Results:
The authors are invited to increase the quality and put all the extensive figures 1-3 in the supplementary material, present in the MS text the obtained values as a mean+/- SD, and show the statistically significant differences.
They are encouraged to ensure that data from figures and tables is accurate and well-presented for both solvents.
5. Discussions:
Please justify the study design, explaining the rationale behind the selection of tumor cell lines and the methods employed.
Please thoroughly discuss the results obtained for both solvents and compare them with those from previously published studies. Please use in this section only numeric values; please do not send the reader to see the figures from the results section.
Moreover, the authors are invited to show the strengths and limitations of the present study.
Then, they could highlight the novelty of their study and its valuable contribution to enrich the scientific database with new and helpful information.
6. Conclusions:
In the current version, this section refers mainly to DMSO. Please revise and briefly present the essential data about both solvents.
Comments on the Quality of English LanguagePlease ensure that all data is presented in the most suitable mode for optimal understanding.
Author Response
The line numbers are missing, so all comments should be made using an approximate localisation.
Line numbers have been included in the manuscript.
- Abstract:
Please organize it in Objective, Materials and Methods, Results, and Conclusions. Please ensure that both solvents are mentioned in all sections. The keywords should include cancer cell lines; please remove 'cancer' and 'cell lines'. MTT assay is the complete name of the test; please correct.
In the revised manuscript, we have reorganized the content into clear sections: background, objective, materials and methods, results, and conclusion. Both solvents, DMSO and ethanol, are now mentioned in each section as advised. Additionally, the keywords section has been revised and now includes cancer cell lines.
- Introduction:
The authors stated: "In drug discovery studies, solvents like Dimethyl Sulfoxide (DMSO) and ethanol are commonly used as carriers or controls [12]. However, these solvents possess intrinsic cytotoxic properties that can confound experimental outcomes by amplifying or mimicking drug effects [13,14]. Moreover, different cell lines exhibit variable sensitivities to these solvents, which can potentially lead to misleading interpretations [15-18].
- However, the authors should mention that most in vitroexperiments use ethanol and DMSO in the cell medium at a concentration < 0.1% and include these solvents as controls to ensure accuracy.
They are invited to find and provide suitable references, as the cytotoxicity of these solvents is well-documented: https://scholar.google.com/scholar?hl=ro&as_sdt=0%2C5&q=dmso+cytotoxicity+on+cells&oq=dmso+cyt
https://scholar.google.com/scholar?hl=ro&as_sdt=0%2C5&q=ethanol+cytotoxicity+on+cells&btnG=
Therefore, the authors are invited to conduct a literature review in the Introduction, providing evidence of the current findings on the cytotoxicity of DMSO and ethanol, as well as the underlying mechanisms involved.
Thank you for your suggestion. We have added a statement to the introduction noting that 0.1% (v/v) DMSO and ethanol in culture medium are commonly used in cell-based assays to facilitate accurate drug screening and have included suitable references.
- Then, they could display the hypotheses for their study and highlight its novelty and valuable contribution to the scientific database.
Thank you for your valuable comment. To our knowledge, this is the first comprehensive study investigating the cytotoxic effects of DMSO and ethanol on this set of cancer cell lines, covering HCC, CRC, and breast cancer, at multiple time points. The hypothesis statement has now been included in the manuscript.
- Materials and Methods
The authors are invited to maintain the original title of this section, because they used both materials and methods, not only methods.
The first subsection should be titled "Materials," where it should include all cell lines and their provenance (complete company name, city, state, and country), reagents, standards, kits for analysis, culture media, consumables, etc.
Then, they should describe the methods used, with suitable references.
Thank you for your comment. The material and methods sections have been organized in the revised manuscript as advised.
- Results:
The authors are invited to increase the quality and put all the extensive figures 1-3 in the supplementary material, present in the MS text the obtained values as a mean+/- SD, and show the statistically significant differences.
Thank you for this valuable comment. All figures have been revised for better clarity. Mean ± SD and statistical significance are now included in the revised results section.
They are encouraged to ensure that data from figures and tables is accurate and well-presented for both solvents.
Thank you for pointing this out. We have revised the data in the figures and tables to ensure accuracy and proper presentation for both solvents.
- Discussions:
Please justify the study design, explaining the rationale behind the selection of tumor cell lines and the methods employed.
Thank you for your comment. We have incorporated a detailed justification for the study design into the revised discussion to clarify the rationale behind our approach.
Please thoroughly discuss the results obtained for both solvents and compare them with those from previously published studies. Please use in this section only numeric values; please do not send the reader to see the figures from the results section.
Thank you for your valuable comment. We included discussions of the data obtained from both solvents and compared them with those from previous studies in the revised manuscript. We also only kept numerical values.
Moreover, the authors are invited to show the strengths and limitations of the present study. Then, they could highlight the novelty of their study and its valuable contribution to enrich the scientific database with new and helpful information.
Thank you for highlighting this point. A paragraph highlighting the strengths, novelty, and limitations of the study has been added at the end of the revised discussion section.
- Conclusions:
In the current version, this section refers mainly to DMSO. Please revise and briefly present the essential data about both solvents.
Thank you for pointing that out. In the revised conclusion, data for both solvents are presented.
Comments on the Quality of English Language
Please ensure that all data is presented in the most suitable mode for optimal understanding.
Thank you for your comment. The manuscript has been carefully checked for grammatical and spelling errors.
Round 2
Reviewer 2 Report
Comments and Suggestions for Authors
The authors replied successfully to all my comments
Author Response
Thank you for your kind feedback. We appreciate your valuable comments, which has helped improve the quality of the manuscript.
Reviewer 3 Report
Comments and Suggestions for Authors
The reviewer highly appreciated the authors' efforts to revise the manuscript in accordance with the previous review report.
Now, the data presented are more clear.
Two minor comments are available:
The authors are encouraged to shorten the big-size phrases to better explain the results (for example, lines 266-272, 106-113).
Moreover, all abbreviations can be placed in a separate table at the end of the manuscript, as the journal template provides.
Comments on the Quality of English Language
The authors are encouraged to shorten the big-size phrases to better explain the results (for example, lines 266-272, 106-113).
Author Response
Thank you for your valuable suggestion. We have implemented a separate abbreviations table at the end of the manuscript, as per the journal template. Additionally, we have revised and shortened the lengthy phrases to improve quality and readability.